# Dynamics of a stochastic SEIQR model driven by Lévy jumps with bilinear incidence rates

**Qiuye Xia, Xiaoling Qiu** *

School of Mathematics and Statistics, Guizhou University, Guiyang, China

* xlqiu@gzu.edu.cn

## Abstract

In this study, we propose a stochastic SEIQR infectious disease model driven by Lévy noise. Firstly, we study the existence and uniqueness of the global positive solution of the model by using the stop-time. Secondly, the asymptotic behavior of the stochastic system at disease-free equilibrium and endemic equilibrium are discussed. Then, the sufficient condition for persistence under the time mean is studied. Finally, our theoretical results are verified by numerical simulation.

**Data Availability Statement:** All relevant data are within the manuscript. (The data source for the graph are in the paragraph above the picture.).

**Funding:** This research was funded by National Natural Science Foundation of China

## Introduction

Infectious diseases have always been one of the important threats to human health, and the control of infectious diseases is an important issue in human society. It is well known that Kermack and McKendrick first proposed the SIR Model based on the Indian plague model [1]. Piovella [2] proposed a SEIR model considering the type E(t) that receives but does not propagate.

Most of the previous models of infectious diseases were basically considered on the basis of assuming the free movement of individuals in the population, and rarely considered the problem of having isolation chambers. With the onset of COVID-19 in 2020, the prevention and treatment of infectious diseases has become one of the topics of research for governments around the world. After the continuous exploration of prevention, the introduction of isolation chamber Q has an excellent effect on timely controlling of infectious diseases. Therefore compared with the previous SIR, SIRS, SEIR and other models, SEIQR model can more accurately describe the prevention and control of infectious diseases. Liu et al. [3] proposed a deterministic SEIQR(S: Susceptible; E: Exposed; I: Infected; Q: Quarantined; R: Removed) epidemic model:

$$
\begin{cases}
dS = (A - \alpha SI - \mu S)dt, \\
dE = [\alpha SI - (a + b)E]dt, \\
dI = [bE - (d + h + \delta)I]dt, \\
dQ = (hI - kQ)dt, \\
dR = (\mu S + aE + dI + kQ - nR)dt,
\end{cases}
\tag{1}
$$

where $t$ is the time; and the normal number $A$ represents the number of births and immigrants entering $S(t)$ per unit time; $\alpha$ is the proportion from $S(t)$ to $E(t)$; $b$ is the incidence of incubation period; $\mu, a, d, k$ represents removal rates from classes $S, E, I, Q$ respectively; $h$ stands for

(No.12061020); Natural Science Foundation of Guizhou Province (QKH[2019]1123, QKH-ZK [2021]331); Natural Science Fundation of Guizhou Province Education Department(QKHKY[2021]088, QKHKY[2022]301); PhD Project of Guizhou Education University(No. 2021BS005). The funders had no role in study design, data collection and analysis, decision to publish, or preparation of the manuscript.

**Competing interests:** The authors have declared that no competing interests exist.

isolation strength; $\delta$ is category $I$ natural mortality; $n$ is the natural mortality rate of category $R$ recoveries due to immune system impairment, age and other reasons.

Converting model (1) to the following form:

$$\frac{dx_i}{dt} = f_i(x) = r_i(x) - h_i(x) = \begin{pmatrix} \alpha SI \\ 0 \\ 0 \\ 0 \\ 0 \end{pmatrix} - \begin{pmatrix} (a+b)E \\ (d+h+\delta)I - bE \\ kQ - hI \\ \alpha SI + \mu S - A \\ nR - \mu S - aE - dI - kQ \end{pmatrix}, i = 1, 2 \ldots m$$

Let $F = \left[\frac{\partial r_i}{\partial x_j}\right]$, $V = \left[\frac{\partial h_i}{\partial x_j}\right]$, where $1 \leq i, j \leq m$, $FV^{-1}$ is called a regenerative matrix. The basic regeneration number is the spectral radius of the regeneration matrix. Basic reproduction number $R_0$ of system (1) is

$$\frac{bk\alpha A}{\mu(a+b)(d+h+\delta)}.$$

If $R_0 \leq 1$, the system (1) has a unique disease-free equilibrium point $P_0 = \left(\frac{A}{\mu}, 0, 0, 0, 0\right)$, and it is locally asymptotically stable; if $R_0 > 1$, system (1) has a unique endemic equilibrium point $P^* = (S^*, E^*, I^*, Q^*, R^*)$, and it is locally asymptotically stable.

Since the infectious disease model is affected by many unpredictable environmental noises, adding random interference to the deterministic model can reflect the transmission law more accurately. In [4–10], white noise interference factor was added to the deterministic model to study the dynamic behavior of a stochastic infectious disease model. Gaussian white noise is used to describe a class of relatively stable, continuous random interference. Tornatore et al. [4] proposed a stochastic SIR model with or without distributed time delay and studied the stability of disease-free equilibrium. Xu et al. [5] studied a kind of SIRS model, proved the existence and uniqueness of the positive solution of the model and obtained the conditions of disease extinction for epidemics. Zhao [6] studied the relationship between the threshold value of stochastic SIRS model with saturation incidence and the extinction and persistence of epidemic diseases. Hieu [7] mainly studied the stochastic SIRS model under telegraph noise and gave the conditions of disease persistence and disease-free equilibrium stability. Cai [8] mainly discussed the limit of transforming SDE model to discrete-time system and proved that the regeneration number can be used to judge the relevant properties of SDE model by using Markov semigroup theory. Yuguo et al. [9] analyzed that the distribution of stochastic SIR model solutions is absolutely continuous. Liu et al. [10] demonstrated that the system has a unique global positive solution and established sufficient conditions for disease persistence. Hattaf et al. [11] proposed and analyzed a stochastic SIR Epidemic model with specific functional response and time delay, and compared the difference of the basic regeneration number between the deterministic model and the stochastic model. Similarly, Lan et al. [12] studied a stochastic SIS model with saturated exposure rates and also found that the conditions for extinction of the disease were much weaker than the corresponding deterministic model. Ali and Khan [13, 14] studied the dynamic properties of stochastic SEIR and SIRS models with saturation rate and simulated them using Legendre spectrum method.

However, disease can be affected by a variety of natural mutations, such as volcanic eruptions, chemical pollutants, and sudden climate changes, which are often not accurately described by stochastic models of Brownian motion. Therefore, many studies on natural mutation factors will use Lévy jump to describe. This perturbation can more accurately describe the impact of mutation factors, and more deeply understand and predict the trend of disease spread and development. According to the Lévy-Itô decomposition theorem [15], Lévy noise is composed of Brownian motion, independent Poisson random measures, and deterministic drift terms, so

Lévy noise has a wider applicability than white noise [16–20]. Zhang and Wang [16, 17] studied SEIR model and S-DI-R model driven by white noise and Lévy noise respectively. Zhang et al. [18] studied the dynamics of a stochastic SIS epidemic model with saturation incidence and dual epidemics and obtained sufficient conditions for the average extinction and persistence of both epidemics. Liu et al. [19] discussed the persistence and extinction of a delayed vaccination SIR epidemic model with temporary immunity and Lévy jumps and analyzed the relationship with the intensity of Lévy noise and the duration of vaccination with the duration of disease and the duration of eradication. EL Koufi et al. [20] considered a stochastic SIR model with a saturated incidence rate and saturated treatment function incorporating Lévy noise. Based on this, a stochastic SEIQR model with Lévy jumps can be considered:

$$
\begin{cases}
dS = (A - \alpha SI - \mu S)dt + \sigma_1 S dB_1(t) + \int_Z C_1(z)S(t)\tilde{N}(dt, dz), \\
dE = [\alpha SI - (a + b)E]dt + \sigma_2 E dB_2(t) + \int_Z C_2(z)E(t)\tilde{N}(dt, dz), \\
dI = [bE - (d + h + \delta)I]dt + \sigma_3 I dB_3(t) + \int_Z C_3(z)I(t)\tilde{N}(dt, dz), \\
dQ = (hI - kQ)dt + \sigma_4 Q dB_4(t) + \int_Z C_4(z)Q(t)\tilde{N}(dt, dz), \\
dR = (\mu S + aE + dI + kQ - nR)dt + \sigma_5 R dB_5(t) + \int_Z C_5(z)R(t)\tilde{N}(dt, dz),
\end{cases}
\tag{2}
$$

where $B_i(t)$ represents standard Brownian motion with filter $\{F_t\}_{t>0}$ on a complete probability space $(\Omega, F, P)$, and they are independent of each other; $\sigma_i > 0$ ($i = 1, 2, 3, 4, 5$) is the intensity of Brownian motion $B_i(t)$; $C_i(Z) > -1$ ($i = 1, 2, 3, 4, 5$) represents the strength of the jump; $N(dt, dz)$ stands for Poisson random measure; $\tilde{N}(dt, dz)$ is the compensating random measure of $N(dt, dz)$, and $\tilde{N}(dt, dz) = N(dt, dz) - \pi(dz)dt$; $\pi(dz)dt$ is the stationary compensator, $\pi$ is a measure defined on a measurable set $Z \subset [0, \infty)$, and satisfies $\pi(Z) < \infty$.

**Lemma 1** (*Itô* formula) If $X(t)$ is the solution of a random differential equation

$$
dx(t) = F(x(t), t)dt + G(x(t), t)dB(t) + \int_Z H(x(t), t, z)\tilde{N}(dt, dz).
$$

If $V \in C^{2,1}(R^d \times [t_0, \infty]; R_+)$, thus the random derivative of $V(x, t)$ is:

$$
dV(x, t) = LV(x, t)dt + V_x(x, t)G(x(t), t)dB(t) + \int_Z [V(x + H(x, t, z)) - V(x, t)]\tilde{N}(dt, dz),
$$

where

$$
\begin{aligned}
LV(x, t) = {}& V_t(x, t) + V_x(x, t)F(x, t) + \frac{1}{2}trace[G^T(x, t)V_{xx}G(x, t)] \\
& + \int_Z [V(x + H(x, t, z)) - V(x, t) - V_x(x, t)H(x, t, z)]\nu(dz).
\end{aligned}
$$

## Existence and uniqueness of the global positive solution

We assume that the jump diffusion coefficient satisfies the following conditions:

(H1): $\int_Z |H_i(x, z) - H_i(y, z)|^2 \pi(dz) \le L_c |x - y|^2$, where $H_1(x, z) = C_1(z)S(t)$, $H_2(x, z) = C_2(z)E$

$(t)$, $H_3(x, z) = C_3(z)I(t)$, $H_4(x, z) = C_4(z)Q(t)$, $H_5(x, z) = C_5(z)R(t)$, $|x| \vee |y| \le c$, where c is a normal number.

(H2): $|C_i(z)| \le K^*$, where $K^*$ is a normal number.

**Lemma 2** Suppose that Conditions (H1) and (H2) hold, for any given initial value $(S(0), E(0), I(0), Q(0), R(0)) \in \mathbb{R}_+^5$, system (2) has a unique positive solution $(S(t), E(t), I(t), Q(t), R(t)) \in \mathbb{R}_+^5$, and the solution lies in $\mathbb{R}_+^5$ with probability 1.

**proof** According to (H1), system (2) satisfies the local Lipschitz condition, for any given initial value, the system (2) has a unique local solution $(S(t), E(t), I(t), Q(t), R(t))$ $(t \in \tau_e)$, where $\tau_e$ represents the blasting time. To prove the existence of a global solution, just prove $\tau_e = \infty$, a.s.

Let $k_0$ be a sufficiently large positive number such that the initial value $(S(0), E(0), I(0), Q(0), R(0))$ is all in $\left[\frac{1}{k_0}, k_0\right]$, For any $k \ge k_0$, the stopping time is defined as:

$$\tau_k = \inf\{t \in [0, \tau_e) : S(t) \notin \left(\tfrac{1}{k}, k\right), E(t) \notin \left(\tfrac{1}{k}, k\right),$$
$$I(t) \notin \left(\tfrac{1}{k}, k\right), Q(t) \notin \left(\tfrac{1}{k}, k\right), R(t) \notin \left(\tfrac{1}{k}, k\right)\}.$$

Obviously, $\tau_k$ is monotonically increasing with respect to $k$ and $\tau_\infty = \lim\limits_{k \to \infty} \tau_k$, thus $\tau_\infty \le \tau_e$. If we could prove that $\tau_\infty = \infty$, then $\tau_e = \infty$.

Next, we use the proof by contradiction to prove. Suppose that $\tau_\infty \ne \infty$, then $\lim\limits_{k \to \infty} \tau_k \ne \infty$, hence there exist constants $N > 0$ and $\varepsilon \in (0, 1)$ such that $P(\tau_\infty \le N) > \varepsilon$.

Then there exists an integer $k_1 \ge k_0$ such that $P(\tau_k \le N) \ge \varepsilon$ for any $k \ge k_1$. Define a $C^2$-function:

$$V = \left(S + n - n\ln\frac{S}{n}\right) + (E + 1 - \ln E) + (I + 1 - \ln I) + (Q + 1 - \ln Q) + (R + 1 - \ln R),$$

where $n$ is a normal number to be determined later. It is easy to judge that $u + 1 - \ln u > 0$ is true, then $V > 0$. And according to Itô formula, we get:

$$
\begin{aligned}
dV &= \frac{\partial V}{\partial S} S\sigma_1 dB_1(t) + \frac{\partial V}{\partial E} E\sigma_2 dB_2(t) + \frac{\partial V}{\partial I} I\sigma_3 dB_3(t) + \frac{\partial V}{\partial Q} Q\sigma_4 dB_4(t) + \frac{\partial V}{\partial R} R\sigma_5 dB_5(t) \\
&\quad + \int_Z [V(S(t) + C_1(z)S(t)) - V(S(t)) + V(E(t) + C_2(z)E(t)) - V(E(t))]\tilde{N}(dt, dz) \\
&\quad + \int_Z [V(I(t) + C_3(z)I(t)) - V(I(t)) + V(Q(t) + C_4(z)Q(t)) - V(Q(t))]\tilde{N}(dt, dz) \\
&\quad + \int_Z [V(R(t) + C_5(z)R(t)) - V(R(t))]\tilde{N}(dt, dz) + LVdt \\
&= LVdt + \left(1 - \frac{n}{S}\right)S\sigma_1 dB_1(t) + \left(1 - \frac{1}{E}\right)E\sigma_2 dB_2(t) + \left(1 - \frac{1}{I}\right)I\sigma_3 dB_3(t) \\
&\quad + \left(1 - \frac{1}{Q}\right)Q\sigma_4 dB_4(t) + \left(1 - \frac{1}{R}\right)R\sigma_5 dB_5(t) + \int_Z [C_1(z)S(t) - n\ln(1 + C_1(z)]\tilde{N}(dt, dz) \\
&\quad + \int_Z [C_2(z)E(t) - \ln(1 + C_2(z)]\tilde{N}(dt, dz) + \int_Z [C_3(z)I(t) - \ln(1 + C_3(z)]\tilde{N}(dt, dz) \\
&\quad + \int_Z [C_4(z)Q(t) - \ln(1 + C_4(z)]\tilde{N}(dt, dz) + \int_Z [C_5(z)R(t) - \ln(1 + C_5(z)]\tilde{N}(dt, dz),
\end{aligned}
\tag{3}
$$

where

$$LV = \left(1 - \frac{n}{S}\right)\frac{dS}{dt} + \left(1 - \frac{1}{E}\right)\frac{dE}{dt} + \left(1 - \frac{1}{I}\right)\frac{dI}{dt} + + \left(1 - \frac{1}{Q}\right)\frac{dQ}{dt} + \left(1 - \frac{1}{R}\right)\frac{dR}{dt}$$

$$+ \frac{1}{2}tr[g^T(t)V_{xx}(t, x(t))g(t)] + n\int_Z [C_1(z) - \ln(1 + C_1(z)]\pi(dz)$$

$$+ \int_Z [C_2(z) - \ln(1 + C_2(z)]\pi(dz) + \int_Z [C_3(z) - \ln(1 + C_3(z)]\pi(dz)$$

$$+ \int_Z [C_4(z) - \ln(1 + C_4(z)]\pi(dz) + \int_Z [C_5(z) - \ln(1 + C_5(z)]\pi(dz).$$

System (2) is substituted into the above formula and the fundamental inequality is used, then

$$LV \leq A + \alpha nI + \mu n + a + b + d + h + k - \delta I + \frac{1}{2}n\sigma_1^2$$

$$+ \sum_{i=2}^{5} \frac{1}{2}\sigma_i^2 + n\int_Z [C_1(z) - \ln(1 + C_1(z)]\pi(dz) + \sum_{i=2}^{5} \int_Z [C_i(z) - \ln(1 + C_i(z)]\pi(dz).$$

Let $n = \frac{\delta}{\alpha}$, then

$$LV \leq A + \mu n + a + b + d + h + k + \frac{1}{2}n\sigma_1^2 +$$

$$+ \sum_{i=2}^{5} \frac{1}{2}\sigma_i^2 + n\int_Z [C_1(z) - \ln(1 + C_1(z)]\pi(dz) + \sum_{i=2}^{5} \int_Z [C_i(z) - \ln(1 + C_i(z)]\pi(dz).$$

According to Taylor's formula and (H2), $|C_i(z) - \ln(1 + C_i(z))| \leq \frac{1}{2}C_i^2(z) \leq \frac{1}{2}K^2*$. Therefore,

$$LV \leq A + \mu n + a + b + d + h + k + \frac{1}{2}n\sigma_1^2 + \sum_{i=2}^{5} \frac{1}{2}\sigma_i^2 + \frac{n+4}{2}K^2* : K$$

Integrating both sides of equation (3) from 0 to $\tau_k \wedge N$ and taking the expectation:

$$E[V(S(\tau_k \wedge N), E(\tau_k \wedge N), I(\tau_k \wedge N), Q(\tau_k \wedge N), R(\tau_k \wedge N))]$$

$$\leq V(S(0), E(0), I(0), Q(0), R(0)) + KN.$$

Let $\Omega_k = \{\tau_k \leq N\}$, then $P(\Omega_k) \geq \varepsilon_0$ for any $k \geq k_1$. Notice that for every $\omega \in \Omega_k$, at least one of $S(\tau_k, \omega), E(\tau_k, \omega), I(\tau_k, \omega), Q(\tau_k, \omega), R(\tau_k, \omega)$ equals either $k$ or $\frac{1}{k}$.

Hence

$$V(S(\tau_k \wedge N), E(\tau_k \wedge N), I(\tau_k \wedge N), Q(\tau_k \wedge N), R(\tau_k \wedge N)) \geq f(k),$$

where $f(k) = \{(k - 1 - \ln k) \wedge (\frac{1}{k} - 1 - \ln\frac{1}{k})\}$, thus

$$V(S(0), E(0), I(0), Q(0), R(0)) + KN$$

$$\geq E[1_{\Omega_k}V(S(\tau_k \wedge T), E(\tau_k \wedge T), I(\tau_k \wedge T), Q(\tau_k \wedge T), R(\tau_k \wedge T))] \geq \varepsilon_0 f(k).$$

Where $1_{\Omega_k}$ is the indicator function of $\Omega_k$, letting $k \to +\infty$, leading to the contradiction:

$$+\infty > V(S(0), E(0), I(0), Q(0), R(0)) + KN = +\infty.$$

Thus $\tau_\infty = \infty$ *a.s.*, then $(S(t), E(t), I(t), Q(t), R(t))$ will not explode in a finite amount of time. Therefore, there is a globally unique positive solution for system (2).

## Asymptotic behavior around disease-free equilibrium of the deterministic model

$P_0 = \left(\frac{A}{\mu}, 0, 0, 0, 0\right)$ is the disease-free equilibrium point of the deterministic model. When $R_0 \leq 1$, $P_0$ is asymptotically stable. Next we will discuss the asymptotic behavior of the solution of the stochastic model at the disease-free equilibrium point.

**Theorem 3** Suppose that conditions (H1), (H2) hold, if $R_0 \leq 1$, and the following conditions are met:

$$\sigma_1^2 + \int_Z C_1^2(z)\pi(dz) < \mu - \frac{n+1}{a_1+1}; \qquad \sigma_2^2 + 2\int_Z C_2^2(z)\pi(dz) < a - n;$$

$$\sigma_3^2 + \int_Z C_3^2(z)\pi(dz)) < \frac{a_2^2 b - 2n}{2a_2}; \qquad \sigma_4^2 + \int_Y C_4^2(z)\pi(dz) < 1 - \frac{2na_3}{ka_2^2 b};$$

$$\frac{1}{2}\sigma_5^2 + \int_Z C_5^2(z)\pi(dz) < \frac{n}{2} - \frac{a^2 + d^2 + k^2}{2n}.$$

For any given initial value $(S(0), E(0), I(0), Q(0), R(0)) \in \mathbb{R}_+^5$, the solution of system (2) has the following properties:

$$\limsup_{t\to\infty} \frac{1}{t} E \int_0^t \left[S(s) - \frac{A}{\mu}\right]^2 + E^2(S) + I^2(S) + Q^2(S) + R^2(S)]ds \leq \frac{M}{\tilde{m}},$$

where

$$m_1 = \mu - \frac{n+1}{a_1+1} - \int_Z C_1^2(z)\pi(dz) - \sigma_1^2; \quad m_2 = a - n - 2\int_Z C_2^2(z)\pi(dz) - \sigma_2^2;$$

$$m_3 = \frac{1}{4}[a_2^2 b - 2a_2(\sigma_3^2 + \int_Z C_3^2(z)\pi(dz)) - 2n]; \quad m_4 = \frac{ka_2^2 b}{4a_3}[1 - (\sigma_4^2 + \int_Z C_4^2(z)\pi(dz))] - \frac{n}{2};$$

$$m_5 = n - \frac{n^2 + a^2 + d^2 + k^2}{2n} - \int_Z C_5^2(z)\pi(dz) - \frac{1}{2}\sigma_5^2; \quad \tilde{m} = \min\{m_1, m_2, m_3, m_4, m_5\};$$

$$M = \frac{A^2}{\mu^2}[(12a_1 + 1)\sigma_1^2 + n + \int_Z (a_1 C_1^2(z) + 1)\pi(dz)].$$

**proof** Define the following functions:

$$V_1 = \frac{1}{2}\left(S - \frac{A}{\mu}\right)^2; V_2 = \frac{1}{2}I^2; V_3 = \frac{1}{2}Q^2; V_4 = E + \frac{a+b}{b}I; V_5 = \frac{1}{2}R^2, V_6 = \frac{1}{2}\left(S - \frac{A}{\mu} + E\right)^2.$$

Thus

$$LV_1 = -\mu\left(S - \frac{A}{\mu}\right)^2 - \alpha\left(S - \frac{A}{\mu}\right)^2 I - \frac{\alpha A}{\mu}\left(S - \frac{A}{\mu}\right)I + \frac{1}{2}\sigma_1^2 S^2 + \frac{1}{2}\int_Z C_1^2(z)S^2(t)\pi(dz). \quad (4)$$

$$LV_2 = [bE - (d + h + \delta)I]I + \frac{1}{2}\sigma_3^2 I^2 + \frac{1}{2}\int_Z C_3^2(z)I^2(t)\pi(dz)$$

$$\leq \frac{b^2}{2(d + h + \delta)}E^2 - \frac{d + h + \delta}{2}I^2 + \frac{1}{2}\sigma_3^2 I^2 + \frac{1}{2}\int_Z C_3^2(z)I^2(t)\pi(dz). \tag{5}$$

$$LV_3 = [hI - kQ]Q + \frac{1}{2}\sigma_4^2 Q^2 + \frac{1}{2}\int_Z C_4^2(z)Q^2(t)\pi(dz)$$

$$\leq \frac{h^2}{2k}I^2 - \frac{1}{2}kQ^2 + \frac{1}{2}\sigma_4^2 Q^2 + \frac{1}{2}\int_Z C_4^2(z)Q^2(t)\pi(dz). \tag{6}$$

$$LV_4 = \alpha\left(S - \frac{A}{\mu}\right)I - \left[\frac{(a + b)(d + h + \delta)}{b} - \frac{\alpha A}{\mu}\right]I \leq \alpha\left(S - \frac{A}{\mu}\right)I. \tag{7}$$

$$LV_5 = R[\mu S + aE + dI + kQ - nR] + \frac{1}{2}\sigma_5^2 R^2 + \frac{1}{2}\int_Z C_5^2(z)R^2(t)\pi(dz)$$

$$\leq \frac{1}{2}n(S^2 + E^2 + I^2 + Q^2) + \left(\frac{\mu^2 + a^2 + d^2 + k^2}{2n} - n\right)R^2 + \frac{1}{2}\int_Z C_5^2(z)R^2(t)\pi(dz). \tag{8}$$

$$LV_6 = \left(S - \frac{A}{\mu} + E\right)(A - \mu S + (a + b)E) + \frac{1}{2}\sigma_1^2 S^2 + \frac{1}{2}\sigma_2^2 E^2$$

$$+ \frac{1}{2}\int_Z C_1^2(z)S^2(t)\pi(dz) + \frac{1}{2}\int_Z C_2^2(z)E^2(t)\pi(dz)$$

$$\leq \left[\frac{(a + b + \mu)^2}{2(a + b)} - \mu + \sigma_1^2\right]\left(S - \frac{A}{\mu}\right)^2 - \frac{1}{2}(a + b)E^2 + \frac{\sigma_1^2 A^2}{\mu^2} + \frac{1}{2}\sigma_2^2 E^2$$

$$+ \frac{1}{2}\int_Z C_1^2(z)S^2(t)\pi(dz) + \frac{1}{2}\int_Z C_2^2(z)E^2(t)\pi(dz). \tag{9}$$

Let $a_1 = \frac{(a+b+\mu)^2}{2(a+b)}$, $a_2 = \frac{d+h+\delta}{b}$, $a_3 = \frac{h^2}{k}$. Define the Lyapunov function again:

$$V = a_1\left(V_1 + \frac{A}{\mu}V_4\right) + a_2\left(V_2 + \frac{a_2 b}{2a_3}V_3\right) + V_5 + V_6.$$

According to the Itô formula, then

$$
\begin{aligned}
dV \quad &= LVdt + a_1\left[\left(S - \frac{A}{\mu}\right)S\sigma_1 dB_1(t) + \frac{A}{\mu}E\sigma_2 dB_2(t) + \frac{A}{\mu}I\sigma_3 dB_3(t)\right] + a_2[I^2\sigma_3 dB_3(t) \\
&\quad + \frac{a_2 b}{2a_3}Q^2\sigma_4 dB_4(t)] + R^2\sigma_5 dB_5(t) + \left(S - \frac{A}{\mu} + E\right)(S\sigma_1 dB_1(t) + E\sigma_2 dB_2(t)) \\
&\quad + a_1\left[\int_Z \left(\left(S - \frac{A}{\mu}\right)C_1(Z)S(t) + \frac{1}{2}(C_1(z)S(t))^2 + \frac{A(a+b)}{\mu b}C_3(z)I(t)\right)\tilde{N}(dt, dz)\right] \\
&\quad + a_2\left[\int_Z C_3(z)I^2(t) + \frac{1}{2}(C_3(z)I(t))^2 + \frac{a_2 b}{2a_3}C_4(z)Q^2(t) + \frac{a_2 b}{4a_3}(C_4(z)Q(t))^2\tilde{N}(dt, dz)\right. \\
&\quad + \int_Z [C_5(z)R^2(t) + \frac{1}{2}(C_5(z)R(t))^2]\tilde{N}(dt, dz) + \int_Z [(S - \frac{A}{\mu} + E)(C_1(z)S(t) + C_2(z)E(t)) \\
&\quad + \frac{1}{2}(C_1(z)S(t) + C_2(z)E(t))^2]\tilde{N}(dt, dz) + \frac{A}{\mu}C_2(z)E(t),
\end{aligned}
\tag{10}
$$

where

$$
\begin{aligned}
LV \quad &= a_1\left(LV_1 + \frac{A}{\mu}LV_4\right) + a_2\left(LV_2 + \frac{a_2 b}{2a_3}LV_3\right) + LV_5 + LV_6 \\
&\leq -[(a_1 + 1)(\mu - \int_Z C_1^2(z)\pi(dz) - \sigma_1^2 - 1) - n - 1]\left(S - \frac{A}{\mu}\right)^2 - [\frac{a - n - \sigma_1^2}{2} \\
&\quad - \int_Z C_2^2(z)\pi(dz)]E^2 - \frac{1}{2}\left[\frac{a_2^2 b}{2} - a_2\sigma_3^2 - a_2\int_Z C_3^2(z)\pi(dz) - n\right]I^2 - \frac{a_2^2 b}{4a_3}\left[k - k\sigma^2\right. \\
&\quad - \int_Z C_4^2(z)\pi(dz) - \frac{a_3 n}{a_2^2 b}\left]Q^2 - \left[n - \frac{\mu + a^2 + d^2 + k^2}{2n} - \frac{1}{2}\sigma_5^2 - \int_Z C_5^2(z)\pi(dz)\right]R^2\right. \\
&\quad + \frac{A^2}{\mu^2}[(12a_1 + 1)\sigma_1^2 + n + \int_Z (a_1 C_1^2(z) + 2)\pi(dz)] \\
&\leq -m_1\left(S - \frac{A}{\mu}\right)^2 - m_2 E^2 - m_3 I^2 - m_4 Q^2 - m_5 R^2 + M.
\end{aligned}
\tag{11}
$$

For the Lyapunov function to be asymptotically stable, then $m_1, m_2, m_3, m_4, m_5 > 0$. Thus $m_1, m_2, m_3, m_4, m_5, \tilde{m}, M$ are described in Theorem 3. Integrating both sides of (10) from 0 to t and taking the expectation:

$$
\begin{aligned}
EV(t) - EV(0) \quad &= E\int_0^t LV(s)ds \\
&\leq E\int_0^t \left[-m_1\left(S - \frac{A}{\mu}\right)^2 - m_2 E^2 - m_3 I^2 - m_4 Q^2 - m_5 R^2\right]ds + Mt.
\end{aligned}
\tag{12}
$$

Then

$$
\lim_{t\to\infty} \sup\frac{1}{t}E\int_0^t \left[S(s) - \frac{A}{\mu}\right]^2 + E^2(s) + I^2(s) + Q^2(s) + R^2(s)]ds \leq \frac{M}{\tilde{m}}.
$$

This completes the proof of Theorem 3.

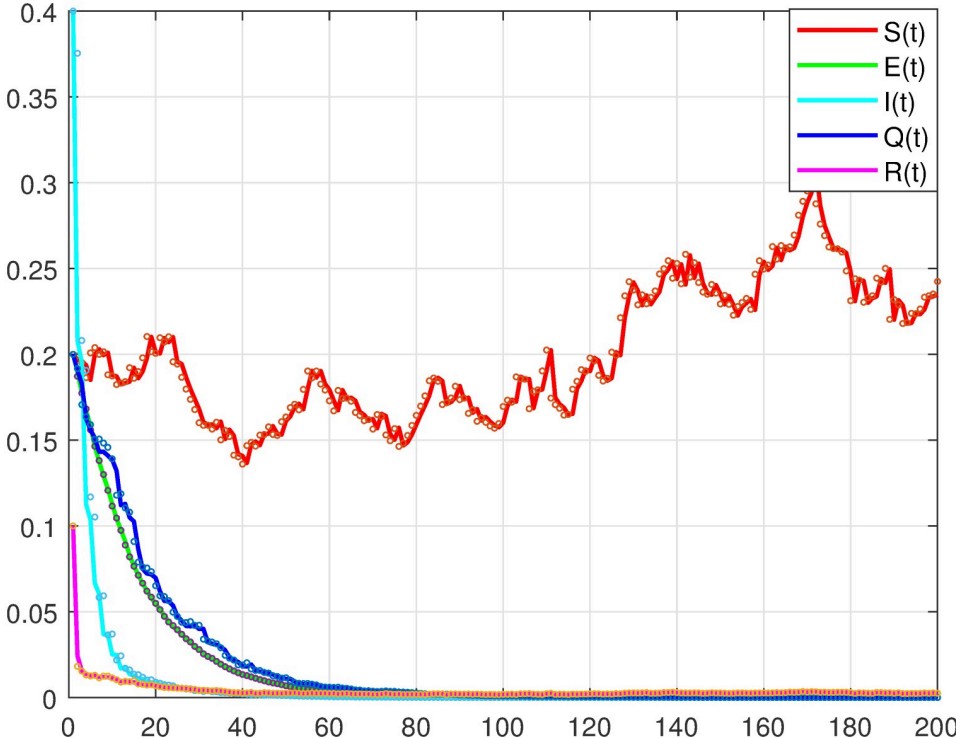

**Fig 1. Asymptotic stability of disease-free equilibrium points in stochastic SEIQR model.**

**Remark 1**: Theorem 3 shows that, with some suitable conditions, the solution of system (2) fluctuates around the disease-free equilibrium point $P_0$, the wave intensity is related to noise intensity $\sigma_i$ and $C_i$. The bigger the $\sigma_i$ and $C_i$, the bigger the fluctuation. That is, the greater the random disturbance, the farther away the solution of system (2) is from the disease-free equilibrium point $P_0$ of the deterministic model, at which time the disease will disappear. Next, we will verify the correctness of Theorem 3 through numerical analysis(see Fig 1, where $A = 0.002$, $\mu = 0.01$, $\sigma_i = 0.03(i = 1, 2, 3, 4, 5)$, $a = 0.0075$, $b = 0.06$, $d = 0.002$, $h = 0.008$, $\alpha = 0.04$, $k = 0.05$, $n = 0.2$).

From the observation of Fig 1, it can be seen that under certain conditions of parameters, the system will stabilize in a situation where only susceptible persons exist. The infected person, exposed person of the virus will disappear, which means the disease will disappear.

## Asymptotic behavior around endemic equilibrium of the deterministic model

$P^*$ is the endemic equilibrium point of the deterministic model. When $R_0 > 1$, $P^*$ is asymptotically stable. Next we will discuss the asymptotic behavior of the solution of the stochastic model at the endemic equilibrium point.

**Theorem 4** Suppose that conditions (H1), (H2) hold, if $R_0 > 1$, and the following conditions are satisfied:

$$\sigma_1^2 + 2\int_Z C_1^2(z)\pi(dz) < \frac{\mu}{2} - \frac{a+b+\mu}{2(a+b)}; \sigma_2^2 + 2\int_Z C_2^2(z)\pi(dz) < \frac{a}{2} - \frac{b^2}{2(d+h+\delta)};$$

$$\left[\sigma_3^2 + \int_Z C_3^2(z)\pi(dz) < \frac{d+h+\delta}{2} - \frac{1}{2\mu}\right]; \qquad \sigma_4^2 + \int_Z C_4^2(z)\pi(dz) < \frac{k}{2} - \frac{1}{2\mu};$$

$$\sigma_5^2 + \int_Z C_5^2(z)\pi(dz) < n - 1 - \frac{b}{2} - \frac{kd^2}{2h^2}.$$

For any given initial value $(S(0), E(0), I(0), Q(0), R(0)) \in \mathbb{R}_+^5$, The solution of system (2) has the following properties:

$$\limsup_{t\to\infty} \frac{1}{t} E \int_0^t \left[(S(s) - S^*)^2 + (E(s) - E^*)^2\right.$$

$$\left. + (I(s) - I^*)^2 + (Q(s) - Q^*)^2 + (R(s) - R^*)^2\right]ds \le \frac{L}{\tilde{l}},$$

where

$$l_1 = \frac{\mu}{2} - \frac{a+b+\mu}{2(a+b)} - \sigma_1^2 - 2\int_Z C_1^2(z)\pi(dz), \qquad l_2 = \frac{a}{2} - \sigma_2^2 - \frac{b^2}{2(d+h+\delta)} - 2\int_Z C_2^2(z)\pi(dz),$$

$$l_3 = \frac{d+h+\delta}{2} - \sigma_3^2 - \frac{1}{2\mu} - \int_Z C_3^2(z)\pi(dz), \qquad l_4 = \frac{k}{2} - \sigma_4^2 - \frac{1}{2\mu} - \int_Z C_4^2(z)\pi(dz),$$

$$l_5 = n - \frac{n^2 + a^2 + d^2 + k^2}{2n} - \int_Z C_5^2(z)\pi(dz) - \frac{1}{2}\sigma_5^2, \qquad \tilde{l} = \min\{l_1, l_2, l_3, l_4, l_5\},$$

$$L = \left[\sigma_1^2 + 2\int_Z C_1^2(z)\pi(dz)\right](S^*)^2 + \left[\sigma_2^2 + 2\int_Z C_2^2(z)\pi(dz)\right](E^*)^2 + \left[\sigma_3^2 + \int_Z C_1^2(z)\pi(dz)\right](I^*)^2$$

$$+ \left[\frac{k}{\mu h^2}\sigma_1^2 + \frac{k}{\mu h^2}\int_Z C_4^2(z)\pi(dz)\right](Q^*)^2 + \left[\sigma_5^2 + \int_Z C_5^2(z)\pi(dz)\right](R^*)^2.$$

**proof** Define the following functions:

$$V_1 = \frac{1}{2}(S - S^* + E - E^*)^2, V_2 = \frac{1}{2}(I - I^*)^2, V_3 = \frac{1}{2}(Q - Q^*)^2,$$

$$V_4 = \frac{1}{2}(R - R^*)^2, V = V_1 + V_2 + \frac{k}{\mu h^2}V_3 + V_4.$$

According to the Itô formula, then

$$
\begin{aligned}
dV &= LVdt + (S - S^* + E - E^*)(\sigma_1 S dB_1(t) + \sigma_2 E dB_2(t)) + (I - I^*)\sigma_3 I dB_3(t) \\
&\quad + \frac{k}{uh^2}(Q - Q^*)\sigma_4 Q dB_4(t) + (R - R^*)\sigma_5 R dB_5(t) \int_Z [(S - S^* + E - E^*)(C_1(z)S(t) \\
&\quad + C_2(z)E(t)) + \frac{1}{2}(C_1(z)S(t) + C_2(z)E(t))^2 + (I - I^*)(C_3(z)I(t)) + \frac{k}{\mu h^2}(Q - Q^*) \\
&\quad \times (C_4(z)Q(t)) + \frac{k}{2\mu h^2}(C_4(z)Q(t))^2 + (R - R^*)C_5(z)R(t) + \frac{1}{2}(C_5(z)R(t))^2]\tilde{N}(dt, dz),
\end{aligned}
\tag{13}
$$

where

$$
\begin{aligned}
LV &= LV_1 + LV_2 + \frac{k}{\mu h^2}LV_3 + LV_4 \\
&\leq \left[\frac{-\mu}{2} + \frac{(a + b + \mu)}{2(a)} + \sigma_1^2 + 2\int_Z C_1^2(z)\pi(dz)\right](S - S^*)^2 + [\sigma_1^2 + 2\int_Z C_1^2(z)\pi(dz)](S^*)^2 \\
&\quad + [\sigma_3^2 + \int_Z C_3^2(z)\pi(dz)](I^*)^2 + [-\frac{a}{2} + \sigma_2^2 + 2\int_Z C_2^2(z)\pi(dz) + \frac{b^2}{2(d + h + \delta)}(E - E^*)^2 \\
&\quad + \left[-\frac{d + h + \delta}{2} + \frac{h^2}{2k} + \sigma_3^2 + \int_Z C_2^2(z)\pi(dz)\right](I - I^*)^2 + [\sigma_5^2 + \int_Z C_4^2(z)\pi(dz)](R^*)^2 \\
&\quad + \frac{k}{\mu h^2}\left[\frac{h - 2k}{2} + \sigma_4^2 + \int_Z C_4^2(z)\pi(dz)\right](Q - Q^*)^2 + \frac{k}{2}(Q - Q^*)^2 \\
&\quad + [1 + \frac{b}{2} + \frac{kd^2}{2h^2} - n + \sigma_5^2 + \int_Z C_5^2(z)\pi(dz)](R - R^*)^2 + [\sigma_2^2 + 2\int_Z C_2^2(z)\pi(dz)](E^*)^2 \\
&\quad + [\frac{k}{\mu h^2}\sigma_1^2 + \frac{k}{\mu h^2}\int_Z C_4^2(z)\pi(dz)](Q^*)^2 \\
&\leq -\left[\frac{\mu}{2} - \frac{a + b + \mu}{2(a + b)} - \sigma_1^2 - 2\int_Z C_1^2(z)\pi(dz)\right](S - S^*)^2 - [\frac{a}{2} - \sigma_2^2 - 2\int_Z C_2^2(z)\pi(dz) \\
&\quad - \frac{b^2}{2(d + h + \delta)}](E - E^*)^2 - \left[\frac{d + h + \delta}{2} - \sigma_3^2 - \int_Z C_3^2(z)\pi(dz) - \frac{1}{2\mu}\right](I - I^*)^2 \\
&\quad - \frac{k}{\mu h^2}\left[\frac{k}{2} - \sigma_4^2 - \int_Z C_3^2(z)\pi(dz) - \frac{1}{2\mu}\right](Q - Q^*)^2 - [n - 1 - \frac{b}{2} - \frac{kd^2}{2h^2} - \sigma_5^2 \\
&\quad - \int_Z C_5^2(z)\pi(dz)](R - R^*)^2 + [\sigma_1^2 + 2\int_Z C_1^2(z)\pi(dz)](S^*)^2 + [\sigma_5^2 + \int_Z C_5^2(z)\pi(dz)](R^*)^2 \\
&\quad + [\sigma_3^2 + \int_Z C_1^2(z)\pi(dz)](I^*)^2 + [\frac{k}{\mu h^2}\sigma_1^2 + \frac{k}{\mu h^2}\int_Z C_4^2(z)\pi(dz)](Q^*)^2 \\
&\quad + [\sigma_2^2 + 2\int_Z C_2^2(z)\pi(dz)](E^*)^2 \\
&\leq -l_1(S - S^*)^2 - l_2(E - E^*)^2 - l_3(I - I^*)^2 - l_4(Q - Q^*)^2 - l_5(R - R^*)^2 + L.
\end{aligned}
\tag{14}
$$

For the Lyapunov function to be asymptotically stable, then $l_1, l_2, l_3, l_4, l_5 > 0$. Where $l_1, l_2, l_3, l_4, l_5, L$ are described in Theorem 4. Integrating both sides of (13) from 0 to t and taking the

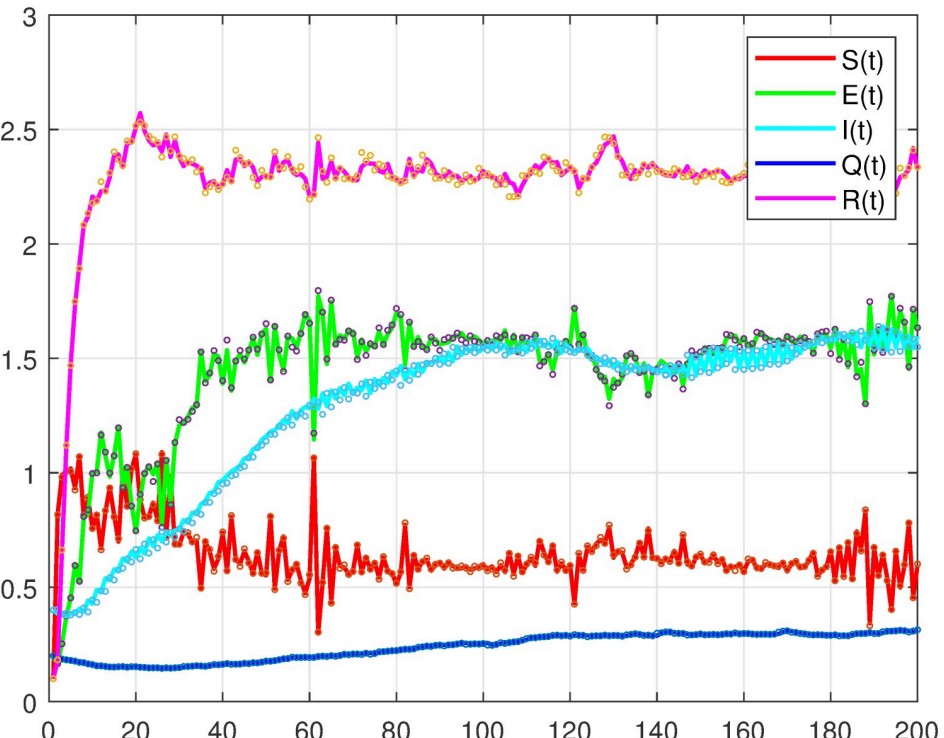

**Fig 2. Asymptotic stability of endemic equilibrium points in stochastic SEIQR model.**

expectation:

$$0 \leq EV(S(t), E(t), I(t), Q(t), R(t)) =$$

$$V(S(0), E(0), I(0), Q(0), R(0)) + E \int_0^t LV(S(\tau), E(\tau), I(\tau), Q(\tau), R(\tau))d\tau. \tag{15}$$

Thus

$$\limsup_{t \to \infty} \frac{1}{t} E \int_0^t \quad [(S(s) - S^*)^2 + (E(s) - E^*)^2$$

$$+ \quad (I(s) - I^*)^2 + (Q(s) - Q^*)^2 + (R(s) - R^*)^2]ds \leq \frac{L}{\tilde{l}},$$

where $\tilde{l}$ is described in Theorem 4.

This theorem is proved.

**Remark 2**: Theorem 4 shows that when some conditions hold, the solution of system (2) oscillates around $P^*$, and the intensity of the vibration is related to the noise intensity. When the degree of disturbance is greater, the solution of the system (2) is further away from the local equilibrium point of the deterministic model, and the disease will persist.

Next, we will verify the correctness of Theorem 4 through numerical analysis(see Fig 2, where $A = 0.8$, $\mu = 0.6$, $\sigma_i = 0.01(i = 1, 2, 3, 4, 5)$, $a = 0.2$, $b = 0.08$, $d = 0.008$, $h = 0.032$, $\alpha = 0.5$, $k = 0.04$, $n = 0.3$).

It can be seen from Fig 2 that, under certain parameter conditions, although the proportion of recovered patients is obvious, the disease will continue because exposed persons and infected persons will still exist in a certain proportion.

## Persistence of disease

**Definition 5** Let $< x(t) >= \frac{\int_0^t x(r)dr}{t}$, if $\liminf_{t \to \infty} < x(t) >> 0$, Then $x(r)$ is called persistence in the sense of time mean.

**Definition 6** If $S(0) + E(0) + I(0) + Q(0) + R(0) \leq \frac{A}{\mu}$, then

$$S(t) + E(t) + I(t) + Q(t) + R(t) \leq \frac{A}{\mu}.$$

Thus $\Omega = \{S(t), E(t), I(t), Q(t), R(t) \in R_+^4 | S(t) + E(t) + I(t) + Q(t) + R(t) \leq \frac{A}{\mu}\}$ is a positive invariant set.

**Theorem 7** Assume that $S(t), E(t), I(t), Q(t), R(t)$ is the system (2) with initial value $(S(0), E(0), I(0), Q(0), R(0)) \in \Omega$ in solution, if $A > \frac{a+b}{\alpha(2a+b)}$, then $\liminf_{t \to \infty} < I(t) >> 0$, the solution $I(t)$ of model (2) is durable in the sense of time mean.

**proof** Let $\Phi(t) = \frac{S(t)-S(0)}{t} + \frac{E(t)-E(0)}{t} + \frac{I(t)-I(0)}{t} + \frac{Q(t)-Q(0)}{t}$. Integrating system (2) from 0 to $t$, dividing by $t$ and substituting $\Phi(t)$ yields:

$$
\begin{aligned}
\Phi(t) \quad &= A - \mu < S(t) > -a < E(t) > -(d+\delta) < I(t) > -k < Q(t) > \\
&+ \frac{1}{t}\int_0^t \sigma_1 S(\tau) + \sigma_2 E(\tau) + \sigma_3 I(\tau) + \sigma_4 Q(\tau)d\tau \\
&+ \frac{1}{t}\int_0^t \int_Z [C_1(z)S(\tau) + C_2(z)E(\tau) + C_3(z)I(\tau) + C_4(z)Q(\tau)]\tilde{N}(dt, dz).
\end{aligned}
\tag{16}
$$

According to the model (2),

$$
\begin{aligned}
\frac{E(t)-E(0)}{t} \quad &= \alpha I(t) < S(t) > -(a+b) < E(t) > \\
&+ \frac{1}{t}\int_0^t \sigma_2 E(\tau)d\tau + \frac{1}{t}\int_0^t \int_Z C_2(z)E(\tau)\tilde{N}(dt, dz).
\end{aligned}
\tag{17}
$$

$$\frac{Q(t)-Q(0)}{t} = h < I(t) > -k < Q(t) > +\frac{1}{t}\int_0^t \sigma_4 Q(\tau)d\tau + \frac{1}{t}\int_0^t \int_Z C_4(z)Q(\tau)\tilde{N}(dt, dz). \tag{18}$$

Therefore

$$
\begin{aligned}
< E(t) > \quad &= \frac{\alpha I(t) < S(t) >}{a+b} - \frac{E(t)-E(0)}{(a+b)t} + \frac{1}{(a+b)t}\Big[\int_0^t \sigma_2 E(\tau)d\tau \\
&+ \frac{1}{t}\int_0^t \int_Z C_2(z)E(\tau)\tilde{N}(dt, dz)\Big].
\end{aligned}
\tag{19}
$$

$$< Q(t) >= \frac{h < I(t) >}{k} - \frac{Q(t)-Q(0)}{kt} + \frac{1}{kt}\Big[\int_0^t \sigma_4 Q(\tau)d\tau + \frac{1}{t}\int_0^t \int_Z C_4(z)Q(\tau)\tilde{N}(dt, dz)\Big]. \tag{20}$$

Let

$$
\begin{aligned}
U &= \frac{1}{t}\int_0^t \sigma_1 S(\tau) + \sigma_2 E(\tau) + \sigma_3 I(\tau) + \sigma_4 Q(\tau)d\tau, \\
V &= \frac{1}{t}\int_0^t \int_Z [C_1(z)S(\tau) + C_2(z)E(\tau) + C_3(z)I(\tau) + C_4(z)Q(\tau)]\tilde{N}(dt,dz).
\end{aligned}
\tag{21}
$$

Substituting (19), (20) into (16):

$$
\begin{aligned}
\Phi(t) &= A - \mu < S(t) > - \frac{a\alpha}{a+b} < E(t) > I(t) - \frac{a}{a+b}\frac{E(t)-E(0)}{t} - (d+\delta+h) < I(t) > \\
&\quad - \frac{Q(t)-Q(0)}{t} - \frac{1}{(a+b)t}\Big[\int_0^t \sigma_2 E(\tau)d\tau + \frac{1}{t}\int_0^t \int_Z C_2(z)E(\tau)\tilde{N}(dt,dz)\Big] \\
&\quad - \frac{1}{kt}\Big[\int_0^t \sigma_4 Q(\tau)d\tau + \frac{1}{t}\int_0^t \int_Z C_4(z)Q(\tau)\tilde{N}(dt,dz)\Big] + U + V.
\end{aligned}
\tag{22}
$$

Thus

$$
\begin{aligned}
(d+h+\delta) < I(t) > &= A - \frac{1}{t}\Big[Q(t) - Q(0) + \int_0^t \sigma_4 Q(\tau)d\tau + \frac{1}{t}\int_0^t \int_Z C_4(z)Q(\tau)\tilde{N}(dt,dz)\Big] \\
&\quad - \frac{a}{(a+b)t}\Big[E(t) - E(0) + \int_0^t \sigma_2 E(\tau)d\tau + \frac{1}{t}\int_0^t \int_Z C_2(z)E(\tau)\tilde{N}(dt,dz)\Big] \\
&\quad + U + V - \Phi(t) + \Big(\mu + \frac{a\alpha}{a+b}\Big) < S(t) > .
\end{aligned}
\tag{23}
$$

According to $\Phi(t) = \frac{S(t)-S(0)}{t} + \frac{E(t)-E(0)}{t} + \frac{I(t)-I(0)}{t} + \frac{Q(t)-Q(0)}{t}$, thus $\lim_{t\to\infty}\Phi(t) = 0$. And according to the strong number theorem:

$$
\begin{aligned}
\lim_{t\to\infty}\frac{1}{t}\int_0^t \sigma_4 Q(\tau)d\tau &= \lim_{t\to\infty}\frac{1}{t}\int_0^t \sigma_2 E(\tau)d\tau = 0, \\
\lim_{t\to\infty}\frac{1}{t}\int_0^t \int_Z C_2(z)E(\tau)\tilde{N}(dt,dz) &= \lim_{t\to\infty}\frac{1}{t}\int_0^t \int_Z C_4(z)Q(\tau)\tilde{N}(dt,dz) = 0.
\end{aligned}
\tag{24}
$$

Because $\Omega$ is positive invariant set, $(S(t), E(t), I(t), Q(t)) \in \Omega$ was founded, that $S(t) + E(t) + I(t) + Q(t) \le \frac{A}{\mu}$, thus $0 \le S(t) \le \frac{A}{\mu}$.

$$
< S(t) > = \frac{1}{t}\int_0^t S(t)d\tau \le \frac{1}{t}\int_0^t \frac{A}{\mu}d\tau < \frac{A}{\mu} + \varepsilon.
\tag{25}
$$

where $\varepsilon$ is any positive constant. In combination with (24) and (25),

$$
\liminf_{t\to\infty} < I(t) > \ge \frac{A}{d+h+\delta} - \frac{\mu^2(a+b) + a\alpha A}{(d+h+\delta)(a+b)\mu}\Big(\frac{A}{\mu} + \varepsilon\Big) > 0.
\tag{26}
$$

This theorem is proved.

Theorem 7 states that under certain conditions, the disease will continue to spread. This means that the disease persists among the population and is not conducive to further management.

## Conclusion

In this work, we have proposed a stochastic SEIQR epidemic model with bilinear incidence rates and Lévy noise based on the randomness of nature and some abrupt fluctuations. By applying the relevant knowledge of stochastic analysis, we have proved the existence and the uniqueness of the global positive solution for the stochastic SEIQR model. Moreover, we showed that the free equilibrium point $P_0$ and the endemic equilibrium point $P^*$ are asymptotically stable under certain conditions. At the same time, we have proved the conditions under which the model is durable in the sense of time mean. Finally, numerical simulation were used to illustrate theoretical results. Different from other three-compartment and four-compartment models, this paper proposes to add isolation compartment and introduce Lévy noise random interference, respectively proving the stability of the equilibrium point and the conditions for the continuous existence of the disease, providing a theoretical basis for the subsequent control of infectious diseases. However, when an infectious disease spreads through a population, the individual gains knowledge about the disease. The classical time derivative cannot reflect the memory effect of model dynamics. The time derivative in this paper is replaced by a fractional derivative [21, 22], and delayed feedback [23] is considered for factors such as vaccines in random infectious diseases. At the same time, we can consider the general non-Markov SEIQR model and compare the discrete and continuous time versions in the future [24].

## Author Contributions

**Conceptualization:** Qiuye Xia.

**Data curation:** Qiuye Xia.

**Formal analysis:** Qiuye Xia.

**Funding acquisition:** Xiaoling Qiu.

**Investigation:** Xiaoling Qiu.

**Methodology:** Qiuye Xia.

**Supervision:** Xiaoling Qiu.

**Writing – original draft:** Qiuye Xia.

**Writing – review & editing:** Qiuye Xia, Xiaoling Qiu.

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
