## [Decision Letter · Decision Letter 0]

15 Apr 2024

PONE-D-24-08895Dynamics of a Stochastic SEIQR Model Driven by L\\'{e}vy Jumps with Bilinear Incidence RatesPLOS ONE

Dear Dr. qiu,

Thank you for submitting your manuscript to PLOS ONE. After careful consideration, we feel that it has merit but does not fully meet PLOS ONE’s publication criteria as it currently stands. Therefore, we invite you to submit a revised version of the manuscript that addresses the points raised during the review process.

We look forward to receiving your revised manuscript.

Kind regards,

Jun Ma, Dr.

Academic Editor

PLOS ONE

Journal Requirements:

"This research was funded by National Natural Science Foundation of China (No.12061020); 

Natural Science Foundation of GuiZhou Province (QKH[2019]1123, QKH-ZK[2021]331); Natural 165

Science Fundation of Guizhou Province Education Department (QKHKY[2021]088,QKHKY[2022]301); 

PhD Project of Guizhou Education University(No. 2021BS005). "

Reviewers' comments:

Reviewer's Responses to Questions

**Comments to the Author**

1. Is the manuscript technically sound, and do the data support the conclusions?

Reviewer #1: Partly

Reviewer #2: Yes

Reviewer #3: Yes

2. Has the statistical analysis been performed appropriately and rigorously? 

Reviewer #1: Yes

Reviewer #2: N/A

Reviewer #3: Yes

3. Have the authors made all data underlying the findings in their manuscript fully available?

Reviewer #1: Yes

Reviewer #2: No

Reviewer #3: Yes

4. Is the manuscript presented in an intelligible fashion and written in standard English?

Reviewer #1: Yes

Reviewer #2: Yes

Reviewer #3: Yes

5. Review Comments to the Author

Reviewer #1: In a revised version the authors should explain this in more details and in simple words and add more physical interpretations, not only refer to theorems and proofs. For more details, please consult my uploaded report. Thank you very much.

Reviewer #2: Review Report

Manuscript ID: PONE-D-24-08895

In this work, a stochastic SEIQR infectious disease model driven by Lévy noise is proposed, considering the impact of discontinuous noise on the transmission process of diseases with latent period. First, the Lyapunov analysis approach is used to demonstrate the existence and uniqueness of the global positive solution of the stochastic SEIQR epidemic model. Then, the asymptotic behavior of stochastic system in the disease-free equilibrium point and endemic equilibrium point is explored by building the Lyapunov function

There is some new contribution and may be consider for publication after addressing the following observations.

1. The abstract is poorly written, it need an improvement, for example, line 2 taking into account may be changed to considering and in line 4, Then, The must be change to Then. the

2. The introduction should also make a compelling case for why the study is useful along with a clear statement of its novelty or originality by providing relevant information and providing answers to basic questions such as:

i. What is already known in the literature?

ii. What was done and how it was done?

3. In introduction section, provide theoretical justification for the choice of Lévy noise over other types of stochastic processes in modeling the abrupt fluctuations inherent in the disease transmission process?

4. In the section on the existence and uniqueness of the global positive solution provide more details on the mathematical intuition and biological implications behind the assumptions (H1) and (H2) related to the jump diffusion coefficient?

5. How did you apply the Ito formula to derive the dynamics of your Lévy-driven stochastic differential equations? A step-by-step mathematical derivation would enhance the clarity. Add detailed mathematical derivations in the subsection discussing the application of the Ito formula.

6. How was the specific form of the Lyapunov function chosen, and what are the mathematical criteria for its selection in proving the asymptotic behavior around the disease-free and endemic equilibrium?

7. Again, in section on the existence and uniqueness of the global positive solution, what mathematical techniques were employed to ensure the solution stays positive and within the biologically feasible region, given the presence of jumps and discontinuities?

8. The conditions for stochastic stability mentioned are quite specific. Provide more insight into how these conditions ensure the stability of the disease-free and endemic equilibrium points? This should be addressed in both the sections on the disease-free equilibrium and the endemic equilibrium analysis.

9. What numerical methods were used for the simulations shown in Figures 1 and 2, and how do these methods accurately capture the effects of Lévy jumps?

10. The references are very few, the author may add some recent related work. They may also add the following recent work, but not mandatory.

https://doi.org/10.3934/math.2023210

https://doi.org/10.1080/07362994.2021.1981382

https://doi.org/10.3390/sym14091838

https://doi.org/10.1007/s11071-021-07095-7

11. Author may look for some punctuation, typos and editing issues.

Reviewer #3: This paper deals with the dynamics of a stochastic SEIQR model driven by Lévy jumps. It can be accepted for the publication after some major revisions.See the attached file. For more details, see the attached file.

6. PLOS authors have the option to publish the peer review history of their article (what does this mean?). If published, this will include your full peer review and any attached files.

Reviewer #1: No

Reviewer #2: No

Reviewer #3: No

---

## [Author Response · Author response to Decision Letter 0]

5 May 2024

At this time, one can see that the quality of this paper has been improved. We express our great thanks again to you. We hope that the corrections and revisions will be satisfactory and that the revised version will be accepted for publication in journal.

---

## [Decision Letter · Decision Letter 1]

27 May 2024

Dynamics of a Stochastic SEIQR Model Driven by L\\'{e}vy Jumps with Bilinear Incidence Rates

PONE-D-24-08895R1

Dear Dr. qiu,

We’re pleased to inform you that your manuscript has been judged scientifically suitable for publication and will be formally accepted for publication once it meets all outstanding technical requirements.

Kind regards,

Jun Ma, Dr.

Academic Editor

PLOS ONE

---

## [Editor Report · Acceptance letter]

4 Jun 2024

PONE-D-24-08895R1 

PLOS ONE

Dear Dr. Qiu, 

I'm pleased to inform you that your manuscript has been deemed suitable for publication in PLOS ONE. Congratulations! Your manuscript is now being handed over to our production team.

Kind regards, 

on behalf of

Dr. and Pro. Jun Ma 

Academic Editor

PLOS ONE